# New BMI Cut-Off Points for Obesity in Middle-Aged and Older Adults in Clinical Nutrition Settings in Italy: A Cross-Sectional Study

**DOI:** 10.3390/nu14224848

**Published:** 2022-11-16

**Authors:** Laura Di Renzo, Leila Itani, Paola Gualtieri, Massimo Pellegrini, Marwan El Ghoch, Antonino De Lorenzo

**Affiliations:** 1Section of Clinical Nutrition and Nutrigenomic, Department of Biomedicine and Prevention, University of Tor Vergata, Via Montpellier 1, 00133 Rome, Italy; 2Department of Nutrition and Dietetics, Faculty of Health Sciences, Beirut Arab University, Riad El Solh, Beirut 1107 2809, Lebanon; 3Department of Biomedical, Metabolic and Neural Sciences, University of Modena and Reggio Emilia, 41125 Modena, Italy

**Keywords:** body composition, BMI, DXA, obesity, body fat, cut-off

## Abstract

Obesity is a major health problem defined as an excess accumulation of body fat (BF). The World Health Organization (WHO) usually relies on a body mass index (BMI) ≥ 30 kg/m^2^ as an indicator of obesity. Due to changes in body composition that occur across the lifespan, with an increase in BF and a decrease in lean mass, we aimed to test the validity of this BMI cut-off point for adiposity in middle-aged and older adults. This cross-sectional study, composed of 4800 adults of mixed gender aged between 40 and 80 years, included (according to the WHO BMI classification) 1087 normal-weight, 1826 overweight, and 1887 obese individuals who were referred to the Department of Biomedicine and Prevention, University of Rome “Tor Vergata”, Italy. The sample was then categorized by adiposity status based on the total BF% as measured by dual-energy X-ray absorptiometry (DXA), and the best sensitivity and specificity were attained for predicting obesity according to the receiver operating characteristic curve (ROC) analysis. In a real-world clinical setting, a new BMI cut-off point (BMI = 27.27 kg/m^2^) has been identified for predicting obesity in middle-aged and older adults. Obesity guidelines in Italy therefore need to be revised accordingly.

## 1. Introduction

Ageing causes changes in body composition [1], such as a decline in lean mass (i.e., muscle), which decreases by approximately almost 5% each decade after the age of 30 [2], and this may lead to an overall reduction of the latter by nearly 20% between the ages of 30 and 70 years [3]. On the other hand, body fat (BF) increases across the lifespan [4], to reach its peak in middle age. The maximum amount of BF has been observed between 50 and 60 years, and that of BF% between 55 and 70 years [5]. These changes usually occur without significant alterations in body weight status or body mass index (BMI) [6]. In this context, obesity become a serious health problem with an increasing prevalence over the past three decades, currently affecting more than 650 million adults worldwide. Furthermore, since it is considered a significant risk factor for several medical [7,8] and psychosocial morbidities [9,10,11,12], as well as increased mortality rates [13], accurate screening for obesity in the early stages is vital [14].

Obesity is best defined as excessive fat deposition in the adipose tissue [15,16]; its identification based on BF quantification seems to be the most accurate method [17]. When classifying obesity in adults, the World Health Organization (WHO) mainly relies on a BMI cut-off point; specifically, a BMI ≥ 30 kg/m^2^ in Caucasians indicates obesity in all age groups of both genders [15]. However, the BMI classification has its limitations [18] since it is not able to discriminate fat from lean mass. Given the changes that occur in body composition across the lifespan, the use of the universal BMI cut-off point (i.e., 30 kg/m^2^ for obesity) for all age groups may not be appropriate and therefore becomes debatable, especially since the relationship between BMI and BF is also age dependent [19]. Recent large sample analysis showed that the WHO BMI cut-off point is not an adequate indicator of obesity in middle-aged and older adults, as it fails to consider changes within the body mass [4]. Secondly, the suggested cut-off point for obesity (i.e., 30 kg/m^2^) is based on observational studies examining the relationship between morbidity and mortality with BMI in certain specific populations (in Europe and the USA). Its validity is not a certainty in other populations, nor has its accuracy been confirmed across different age, gender, or ethnic groups [19,20,21,22,23,24]. Moreover, in Italy, agreement between BMI and BF categories is low for the total population, and rare for females [25]. Therefore, a new predictive equation for BF was developed, which is easily applicable to Italian women [26].

Based on these considerations, the current study aims to investigate to what extent the WHO BMI cut-off point for obesity classification (i.e., 30 kg/m^2^) is accurate in a clinical nutrition setting composed of middle-aged and older patients in Italy and, if not, to identify more suitable ones wherever necessary.

## 2. Materials and Methods

### 2.1. Participants and Design of the Study

This single-center cross-sectional observational study was conducted in adherence to the strengthening the reporting of observational studies in epidemiology (STROBE) guidelines (STROBE checklist in Appendix A) [27]. The participants were selected from a large pool of patients whom general practitioners referred and subsequently enrolled in the Division of Clinical Nutrition at the Department of Biomedicine and Prevention, University of Rome “Tor Vergata”, Italy, during the period between June 2018 and May 2022. Patients were considered eligible to participate if they were adults and had completed a body composition measurement using dual-energy X-ray absorptiometry (DXA). A total of 8722 patients were considered eligible and were checked for inclusion and exclusion criteria. The inclusion criteria included being between 40 and 80 years old and therefore defined as middle aged or elderly, with a BMI ≥ 20 kg/m^2^. Patients were excluded if they were pregnant (among females), taking medication that affects body weight or composition, or presented with medical comorbidities associated with weight loss (i.e., cancers) or severe psychiatric disorders at the baseline assessment. A total of 4800 individuals of different genders and different body weight statuses according to the WHO BMI classification—normal weight (*n* = 1087), overweight (*n* = 1826), or obesity (*n* = 1887)—were included. The research was conducted in accordance with the Declaration of Helsinki and was approved by the Ethics Committee of the Calabria Region Center Area Section (Register Protocol No. 146 17/05/2018). All patients’ personal data were treated according to European/Italian privacy laws, and informed written consent was obtained.

### 2.2. Body Weight and Height

Body weight and height were measured with the participants wearing light clothes and no shoes, using an electronic weighing scale (SECA 2730-ASTRA, Hamburg, Germany) and a stadiometer. BMI was then calculated according to the standard formula of body weight measured in kilograms divided by the square of the height in meters.

### 2.3. Body Composition

Body composition was determined using a DXA (Primus, X-ray densitometer; software version 1.2.2, Osteosys Co., Ltd., Guro-gu, Seoul, Republic of Korea) fan beam scanner. It assessed both whole and segmental composition regarding fat and lean mass. Patients were given complete and standardized instructions on the testing procedure, as described elsewhere [25]. The sample was categorized according to age- and gender-specific obesity cut-off points as follows [21]:40–59 years: BF% ≥ 40% for females and BF% ≥ 28% for males.60–79 years: BF% ≥ 42% for females and BF% ≥ 30% for males.

### 2.4. Statistical Analysis

Descriptive statistics are presented as means and standard deviations for continuous variables and frequencies, with proportions for categorical variables. Pearson’s correlation coefficient and scatter plots were used to assess the association between BF% and BMI. A cumulative sum (CUSUM) linearity test was conducted to confirm a linear association between BMI and BF% [28]. A *p*-value > 0.05 for the CUSUM test indicated a positive linear association. To evaluate the diagnostic performance of BMI in detecting obesity status defined by BF% for all subjects by age and gender, a classification analysis was performed by calculating sensitivity and specificity as well as the area under the curve (AUC) of the receiver operating characteristic curve (ROC). For this purpose, the definition of obesity based on age- and gender-specific BF% as indicated above was used as a gold standard [21]. The criterion value of BMI with maximum sensitivity and specificity was selected for the age- and gender-specific BMI cut-off points. An AUC > 0.8 indicates that the criterion value has an excellent discriminating ability [29]. All values were considered significant at *p* < 0.05. The Number Cruncher Statistical Systems (NCSS) 12.0.2 (NCSS, NCSS, Kaysville, UT, USA) package was used for the statistical analysis. A post-hoc Power analysis for the sample size was determined with Power Analysis and Sample Size (PASS) software (PASS 11, NCSS, Kaysville, UT, USA). For the sample of 3194 patients classified as obese versus 1606 categorized as not, with an alpha of 0.05 and AUC of 0.88, the power was 1.000.

## 3. Results

The study sample comprised 1850 (38.5%) males and 2950 (61.5%) females, with a mean age of 54.8 ± 9.7 years and 54.0 ± 9.1 years, respectively. The mean BMI was 29.3 ± 4.5 kg/m^2^ in males and 28.8 ± 4.9 kg/m^2^ in females, with almost 40% of both groups categorized as obese (38.4% vs. 40.6%) based on the WHO criteria. However, according to the classification based on BF%, almost two-thirds of the males (71.1%) and females (63.6%) were obese (Table 1).

The correlation analysis revealed a significant positive correlation between BMI and BF% in the overall sample of males (ρ = 0.709, *p*-value < 0.0001) and females (ρ = 0.741, *p*-value < 0.0001). A significant positive linear association between BMI and BF% (Figure 1a–f) among male and female participants aged 40–59 years and those aged 60–79 years, as well as the total sample (40–79 years), was confirmed by the CUSUM linearity test (*p* > 0.05).

The results of the ROC analysis on the diagnostic performance of BMI by age group and gender are shown in Table 2 and Figure 2a–i. The females in age group 1 (40–59 years) had a mean age of 49.8 ± 5.5 years and a mean BMI of 28.6 ± 4.9 kg/m^2^, with one-third either overweight (34.7%) or obese (37.5%) based on the WHO criteria. Alternatively, almost two-thirds (64.0%) of females in this age group were categorized as obese according to BF% classification (Table 1). The ROC analysis among females aged 40–59 showed that the most appropriate BMI cut-off point for identifying obesity based on BF% was 27.03 kg/m^2^. The BMI cut-off point for this group achieved high sensitivity (80.69%) and specificity (83.63%), indicating a low chance of false negatives and false positives (Table 2). A comparison of the distribution of obesity determined by BF and the WHO criteria revealed a disagreement, with only 56.2% of those defined as obese based on BF% being correctly classified according to the WHO criteria, thus missing almost half of those who are obese (Table 3). After comparing the proportion correctly diagnosed by the WHO cut-off point (56.2%), the identified cut-off for females aged 40–59 years improved the detection of obesity by nearly 25% (24.5%), reaching 80.7% (Table 3). The AUC (0.90) depicts the excellent discriminating ability of BMI, which has a 90% chance of detecting obesity (Table 2, Figure 2a).

The females in age group 2, ranging between 60 and 79 years, had a mean age of 66.6 ± 5.2 years and a mean BMI of 29.4 ± 4.9 kg/m^2^, with 40% being overweight (40.8%) or obese (41.0%) based on the WHO classification criteria. Alternatively, almost two-thirds (62.3%) of females in this age group were identified as obese based on BF% (Table 1). The ROC analysis among females aged 60–79 demonstrated that the most appropriate BMI cut-off point for diagnosing obesity based on BF% was 27.11 kg/m^2^. The BMI cut-off point for this group achieved high sensitivity (85.06%) and relatively lower specificity (74.29%), indicating a low chance of false negatives and false positives (Table 2). A comparison of the distribution of obesity determined by BF and the WHO criteria revealed a disagreement, with only 59.5% of those defined as obese based on BF% being correctly classified according to the WHO criteria, thus missing almost 40% of those who are obese (Table 3). After comparing the proportion correctly diagnosed by the WHO cut-off point (59.5%), the identified cut-off for females aged 60–79 improved the detection of obesity based on BF by 25% (25.3%), reaching 84.8% (Table 3). The AUC (0.87) indicates the excellent discriminating ability of BMI, which has an 87% chance of detecting obesity (Table 2, Figure 2b).

The age of the whole female sample ranged between 40 and 79 years, with a mean age of 54.0 ± 9.1 years and a mean BMI of 28.8 ± 4.9 kg/m^2^, with one-third either being overweight (36.3%) or obese (38.4%) according to the WHO classification criteria. Alternatively, almost two-thirds (63.6%) of females in this age group were identified as obese based on BF% (Table 1). The ROC analysis among females aged 40–79 demonstrated that the most appropriate BMI cut-off point for diagnosing obesity according to BF% was 27.08 kg/m^2^. The BMI cut-off point for this group achieved high sensitivity (81.66%) and specificity (81.19%), indicating a low chance of false negatives and false positives (Table 2). Comparing the distribution of obesity determined by BF and the WHO criteria revealed a disagreement, with only 57.0% of those defined as obese based on BF% being correctly classified according to the WHO criteria, thus missing almost 40% of those who are obese (Table 3). After comparing the proportion correctly diagnosed by the WHO cut-off point (57.0%), the identified cut-off for females aged 40–79 years improved the detection of obesity according to BF by 25% (24.7%), reaching 81.7% (Table 3). The AUC (0.89) illustrates the excellent discriminating ability of BMI, which has an 89% chance of detecting obesity (Table 2, Figure 2c).

The males in age group 1, ranging between 40 and 59 years, had a mean age of 49.5 ± 5.7 years and a mean BMI of 29.3 ± 4.5 kg/m^2^, with 40% being overweight (40.5%) or obese (40.6%) according to the WHO criteria. Alternatively, over two-thirds (71.7%) of males in this age group were identified as being obese based on BF (Table 1). The ROC analysis among males aged 40–59 demonstrated that the most appropriate BMI cut-off point for diagnosing obesity according to BF% was 27.36 kg/m^2^. The BMI cut-off point for this group achieved high sensitivity (80.0%) and good specificity (79.40%), indicating a low chance of false negatives and false positives (Table 2). Comparing the distribution of obesity determined by BF and the WHO criteria revealed a disagreement. Only 54.4% of those defined as obese based on BF% were correctly classified according to the WHO criteria, thus missing almost half of those who are obese (Table 3). When comparing the proportion correctly diagnosed by the WHO cut-off point (54.4%), the identified cut-off for males aged 40–59 improved the detection of obesity based on BF by 25% (25.6%), reaching 80.0% (Table 3). The AUC (0.88) indicates the excellent discriminating ability of BMI, which has an 88% chance of detecting obesity (Table 2, Figure 2d).

The males in age group 2, ranging between 60 and 79 years, had a mean age of 66.8 ± 5.1 years and a mean BMI of 29.2 ± 4.2 kg/m^2^, with 40% being overweight (41.8%) or obese (41.1%) based on the WHO classification criteria. Alternatively, more than two-thirds (70.3%) of males in this age group were categorized as obese according to BF% (Table 1). The ROC analysis among males aged 60–79 showed that the most appropriate BMI cut-off point for diagnosing obesity defined by BF% was 27.25 kg/m^2^. The BMI cut-off point for this group achieved high sensitivity (81.11%) and relatively low specificity (74.40%), indicating a low chance of false negatives and a relatively low chance of false positives (Table 2). Comparing the distribution of obesity determined by BF% and the WHO criteria revealed a disagreement, with only 55.4% of those defined as obese based on BF% being correctly classified according to the WHO criteria, thus missing almost half of those who are obese (Table 3). After comparing the proportion correctly diagnosed by the WHO cut-off point (55.4%), the identified cut-off for males aged 60–79 improved the detection of obesity according to BF% by 25% (25.7%), reaching 81.1% (Table 3). The AUC (0.85) indicates the excellent discriminating ability of BMI, which has an 85% chance of detecting obesity (Table 2, Figure 2e).

The age of the whole male sample ranged between 40 and 79 years, with a mean age of 54.8 ± 9.7 years and a mean BMI of 29.3 ± 4.4 kg/m^2^, with 40% either being overweight (40.9%) or obese (40.8%) according to the WHO classification criteria. Alternatively, almost two-thirds (71.2%) of males in this age group were classified as obese based on BF% (Table 1). The ROC analysis among males aged 40–79 showed that the most appropriate BMI cut-off point for diagnosing obesity based on BF% was 27.36 kg/m^2^. The BMI cut-off point for this group achieved high sensitivity (80.00%) and specificity (79.40%), indicating a low chance of false negatives and false positives (Table 2). Comparing the distribution of obesity determined by BF% and the WHO criteria revealed a disagreement, with only 54.7% of those defined as obese according to BF% being correctly categorized according to the WHO criteria, thus missing almost half of those who are obese (Table 3). After comparing the proportion correctly diagnosed by the WHO cut-off point (54.7%), the identified cut-off for males aged 40–79 years improved the detection of obesity according to BF by almost 25% (24.4%), reaching 80.0% (Table 3). The AUC (0.88) indicates the excellent discriminating ability of BMI, which has an 88% chance of detecting obesity (Table 2, Figure 2f).

Finally, considering the sensitivity and specificity of the WHO cut-off point for defining obesity (≥30 kg/m^2^), overall, this cut-off point demonstrated poor sensitivity, reaching less than 60%, and high specificity, achieving more than 90% across all age and sex groups. Therefore, it has a higher chance of false positives and false negatives (Table 3).

## 4. Discussion

The current study aimed to provide benchmark data regarding the validity of the WHO BMI cut-off point of 30 kg/m^2^ for obesity in middle-aged and older patients of both genders and, in the case of non-validity, to determine a new BMI cut-off point for more accurate screening for obesity in a clinical nutrition setting in Italy.

### 4.1. Findings and Concordance with Previous Studies

Our main finding was the identification of a new BMI cut-off point for obesity in a mainly clinical sample of participants enrolled in a nutritional management setting. This new cut-off point (i.e., BMI = 27.27 kg/m^2^) in our population was found to be significantly lower than the WHO BMI cut-off point (i.e., BMI = 30 kg/m^2^) [16]. Even though we are not in a position to determine the exact reason behind this discrepancy between the BMI cut-off for obesity in our population and that suggested by the WHO, we can, however, speculate and suggest that the changes in body composition across one’s lifespan (i.e., the increased BF and reduction in lean mass), which seem to occur without a relevant variation in body weight, lead to higher adiposity at a lower BMI. Our investigation is not the first to suggest lower cut-off points for identifying obesity in middle-aged and older populations [30,31]. Other studies have suggested a BMI of approximately 27 kg/m^2^ as a cut-off point for obesity in Western people [32,33]. For instance, Evans et al. identified a BMI > 27.0 kg/m^2^ as defining obesity in white, middle-aged women in the United States [32]. Fernández-Real et al. recognized a BMI > 27.5 kg/m^2^ (more specifically, 27.5 kg/m^2^ in males and 27.4 kg/m^2^ in females) as being indicative of obesity in the Spanish population [33]. The populations studied in the aforementioned studies are similar to a certain extent to our population in terms of several aspects (i.e., age, ethnicity, etc.) [32,33]. Outside of Western populations, Jahanlou and Kouzekanani detected obesity among Iranian adults and found that the appropriate cut-off points for BMI to define obesity were 27.70 kg/m^2^ for females and 27.30 kg/m^2^ for males [22], which overlap with our cut-off points.

### 4.2. Potential Clinical Implications and New Directions

Our findings have several clinical implications. Firstly, policy-makers in Italy are invited to take our results as at least preliminary evidence for considering this cut-off point as a new one for identifying obesity in clinical settings. Second, awareness should be raised among all healthcare professionals dealing with obesity with regard to recognizing this new cut-off point when screening for obesity and sharing or discussing this new information with their patients.

In addition, some new directions for future research are needed. Firstly, other investigations should replicate our findings to confirm this cut-off point (i.e., 27.27 kg/m^2^) in middle-aged and older Italians. Moreover, future investigations are also needed to determine the cut-off point for leanness (i.e., underweight), especially in older adults, since previous research has shown that adults over 60–65 years who are underweight experience more health issues [34,35] and shorter life expectancy [35,36] than those with higher BMI (i.e., <27 kg/m^2^) [37]. Secondly, on a national level, large sample studies must also assess the validity of normal (≥18.5 kg/m^2^) and overweight (≥25 kg/m^2^) cut-off points and, if necessary, determine new ones in both clinical and general populations. Finally, other works should extend the aim of our analysis to other European countries (i.e., Central, Eastern, Northern, Southern, and Western Europe).

### 4.3. Study Strengths and Limitations

Our study has several strengths. To the best of our knowledge, it is one of the few evaluations, if not the first, to test the validity of the WHO BMI cut-off point (i.e., 30 kg/m^2^) and determine a new one for obesity in a large group of patients composed of middle-aged and older adults in a “real-world” clinical setting in Italy. Second, body composition was measured using DXA, which is known to exhibit a high level of precision [38]. However, our investigation also had some limitations. Most importantly, the data were obtained from a single unit, so external validation in other populations is necessary [39]. Finally, our study was of cross-sectional design; therefore, it was unable to detect BMI trends or changes (i.e., during COVID pandemic), which usually requires longitudinal assessment [40].

## 5. Conclusions

Obesity is a chronic disease associated with other severe comorbidities over time. Early identification is, therefore, crucial for managing the progression of the latter [41]. In our study, we provide evidence that the optimal BMI cut-off point (i.e., ≥27 kg/m^2^) corresponding to obesity in a middle-aged and older mixed-gender clinical population in Italy varies from the widely used one (i.e., ≥30 kg/m^2^). Therefore, we recommend that this new cut-off point be applied in clinical settings when screening individuals for obesity in Italy.

## Figures and Tables

**Figure 1 nutrients-14-04848-f001:**
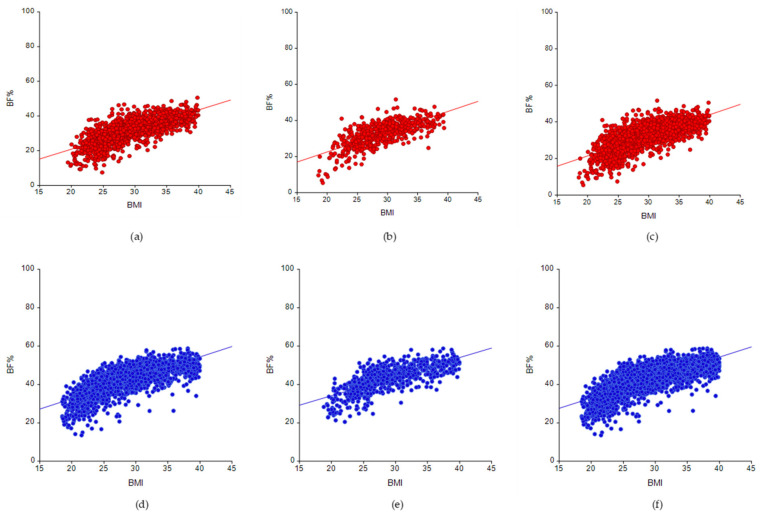
BMI versus BF% by gender and age groups (BF—body fat percentage; BMI—body mass index). (**a**) Males (40–59 years), (**b**) Males (60–79 years), (**c**) Males (40–79 years), (**d**) Females (40–59 years), (**e**) Females (60–79 years), (**f**) Females (40–79 years).

**Figure 2 nutrients-14-04848-f002:**
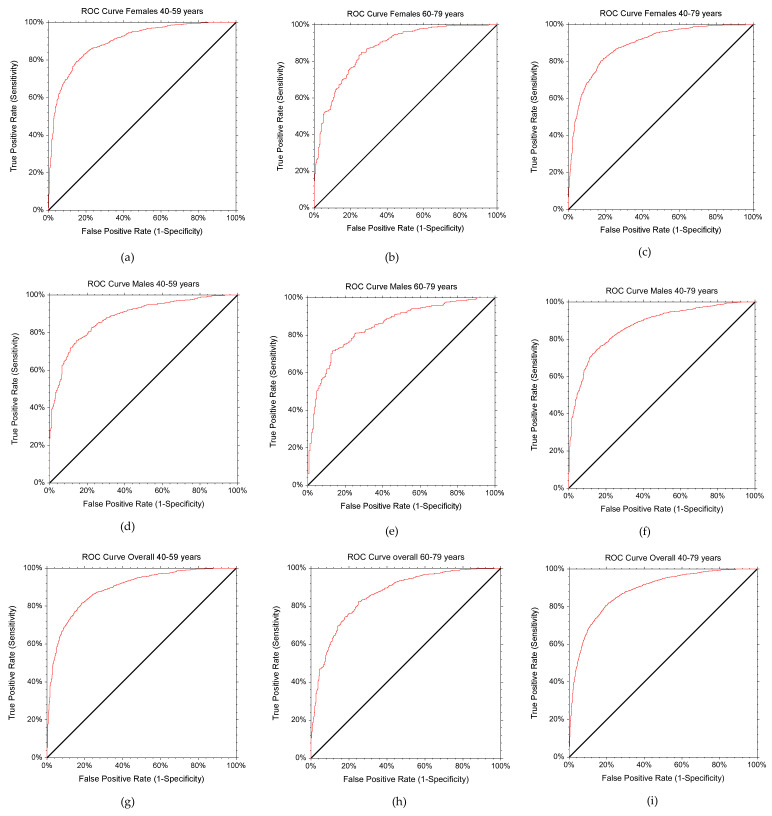
Receiver operator characteristic curve by age and gender for BMI cut-off point to detect obesity based on BF%. (**a**) Females 40–59 years, (**b**) Females 60–79 years, (**c**) Females 40–79 years, (**d**) Males 40–59 years, (**e**) Males 60–79 years, (**f**) Males 40–79 years, (**g**) Overall 40–59 years, (**h**) Overall 60–79 years, (**i**) Overall 40–79 years.

**Table 1 nutrients-14-04848-t001:** Anthropometric and body composition characteristics of the study sample (*n* = 4800) ^Ŧ^.

Variable	Female			Males		
	Total N = 2950	40–59 N = 2208	60–79 N = 742	Total N = 1850	40–59 N = 1285	60–79 N = 565
Age (years)	54.0 (9.1)	49.8 (5.5)	66.6 (5.2)	54.8 (9.7)	49.5 (5.7)	66.8 (5.1)
Weight (kg)	73.3 (13.2)	73.7 (13.3)	72.0 (12.8)	87.9 (15.0)	89.5 (15.2)	84.3 (13.9)
Height (cm)	159.5 (6.5)	160.6 (6.2)	156.5 (6.5)	173.2 (7.0)	174.7 (6.5)	169.8 (6.9)
BMI (kg/m^2^)	28.8 (4.9)	28.6 (4.9)	29.4 (4.9)	29.3 (4.4)	29.3 (4.5)	29.2 (4.2)
WHO classification ^£^						
Normal weight	747 (25.3)	612 (27.7)	135 (18.2)	340 (18.4)	243 (18.9)	97 (17.2)
Overweight	1070 (36.3)	767 (34.7)	303 (40.8)	756 (40.9)	520 (40.5)	236 (41.8)
Obesity	1133 (38.4)	829 (37.5)	304 (41.0)	754 (40.8)	522 (40.6)	232 (41.1)
BF (kg)	31.7 (9.5)	31.7 (9.6)	31.9 (9.3)	28.7 (9.6)	28.9 (10.0)	28.3 (8.7)
BF (%)	42.5 (6.7)	42.2 (6.7)	43.6 (6.4)	32.0 (6.7)	31.5 (6.9)	33.0 (6.2)
BF Classification ^¥^						
Normal weight	362 (12.3)	269 (12.2)	93 (12.5)	185 (10.0)	125 (9.7)	60 (10.6)
Overweight	712 (24.1)	525 (23.8)	187 (25.2)	347 (18.8)	239 (18.6)	108 (19.1)
Obese	1876 (63.6)	1414 (64.0)	462 (62.3)	1318 (71.2)	921 (71.7)	397 (70.3)
Lean tissue mass (kg)	39.2 (5.5)	39.6 (5.6)	38.0 (5.2)	56.2 (7.6)	57.5 (7.5)	53.1 (7.0)
Lean Tissue mass (%)	54.2 (6.4)	54.5 (6.4)	53.4 (6.2)	64.5 (6.4)	65.0 (6.5)	63.5 (5.9)

^Ŧ^ Values are presented as means (SD) for continuous variables and n (%) for categorical variables. BMI—body mass index; BF—body fat. ^£^ WHO classification for BMI. ^¥^ Age- and gender-specific obesity cut-off points according to BF%.

**Table 2 nutrients-14-04848-t002:** Diagnostic performance of the new BMI cut-off points for obesity by age and gender and overall sample (*n* = 4800).

	Total	Obese Subjects	BMI Cut-Off Point for Obesity	AUC (95% CI)	*p*-Value	Sensitivity	Specificity
Total group (40–79 years)	4800	3194	27.27	0.88 (0.87–0.89)	<0.0001	0.8040	0.8076
Females	2950	1876	27.08	0.89 (0.88–0.90)	<0.0001	0.8166	0.8119
Males	1850	1318	27.36	0.88 (0.86–0.90)	<0.0001	0.8000	0.7940
Age group 1 (40–59 years)	3439	2335	27.08	0.89 (0.88–0.91)	<0.0001	0.8141	0.8178
Females	2208	1414	27.03	0.90 (0.88–0.91)	<0.0001	0.8069	0.8363
Males	1285	921	27.36	0.88 (0.86–0.90)	<0.0001	0.8000	0.7940
Age group 2 (60–79 years)	1307	859	27.26	0.86 (0.84–0.88)	<0.0001	0.8219	0.7522
Females	462	742	27.11	0.87 (0.84–0.90)	<0.0001	0.8506	0.7429
Males	565	397	27.25	0.85 (0.82–0.89)	<0.0001	0.8111	0.7440

**Table 3 nutrients-14-04848-t003:** Proportion correctly diagnosed by WHO cut-off points and new cut-off points and the sensitivity and specificity at BMI ≥ 30 kg/m^2^.

Age and Gender Distribution	Proportion Correctly Diagnosed		BMI ≥ 30 kg/m^2^	
	WHO Cut-Off Point	New Cut-Off Point	Sensitivity	Specificity
Total group (40–79 years)	56.0	80.4	56.04	93.96
Females	57.0	81.7	57.09	94.36
Males	54.7	80.0	54.70	93.80
Age group 1 (40–59 years)	55.5	81.4	55.55	95.16
Females	56.2	80.7	56.29	95.59
Males	54.4	80.0	54.40	94.23
Age group 2 (60–79 years)	57.6	82.2	57.63	90.85
Females	59.5	84.8	59.74	89.64
Males	55.4	81.1	55.42	92.86

## Data Availability

The dataset in the present study is available upon request.

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
