# Peer review of "New BMI Cut-Off Points for Obesity in Middle-Aged and Older Adults in Clinical Nutrition Settings in Italy: A Cross-Sectional Study"

_nutrients, 2022, doi:10.3390/nu14224848_

Round 1
Reviewer 1 Report
This study includes a valuable purpose for public health, and the results may contribute to the development of obesity epidemiology.
However, there are serious problems, and the manuscript needs to make corrections for the reviewing and proofreading of it. In particular, the following points should be revised.
1) Previous studies have shown that being thin has an impact on the health of older people than being obese. This study does not allow for these discussions because it excluded lean people. Please add discussions on this point and, if necessary, perform additional analyses of all participants, including lean people.
2) During COVID-19 pandemic, lack of exercise and disturbed eating habits have been reported from many studies. Was there a difference in people's BMI trends before and during the pandemic in this study? Whether it was affected or not, the impact of the pandemic should be mentioned by the authors.
3) In this study participants, were pregnant females excluded? (or was it not included from the beginning of this study?)
4) After revising the above points, check the STROBE guideline for cross-sectional study and submit the checklist. (https://www.strobe-statement.org/checklists/).
Author Response
This study includes a valuable purpose for public health, and the results may contribute to the development of obesity epidemiology. However, there are serious problems, and the manuscript needs to make corrections for the reviewing and proofreading of it. In particular, the following points should be revised.
1) Previous studies have shown that being thin has an impact on the health of older people than being obese. This study does not allow for these discussions because it excluded lean people. Please add discussions on this point and, if necessary, perform additional analyses of all participants, including lean people.
Response: We thank the reviewer for the comment. Now we add a small paragraph on the raised point in the Discussion section under the new directions for future research (Page 12, paragraph 2).
2) During COVID-19 pandemic, lack of exercise and disturbed eating habits have been reported from many studies. Was there a difference in people's BMI trends before and during the pandemic in this study? Whether it was affected or not, the impact of the pandemic should be mentioned by the authors.
Response: our study is of cross-sectional design therefore it was unable to detect the BMI trends/changes (i.e. during COVID pandemic), which requires longitudinal assessment. Moreover it does not fall within our main aim. However we mentioned this clearly as limitation of the study (Page 12, paragraph 3), and added suitable reference.
3) In this study participants, were pregnant females excluded? (or was it not included from the beginning of this study?)
Response: Pregnancy was an exclusion criterion. Therefore pregnant females were not included from the beginning (Page 2, paragraph 3).
4) After revising the above points, check the STROBE guideline for cross-sectional study and submit the checklist. (https://www.strobe-statement.org/checklists/).
Response: Done as suggested, reported in the Method section (Page 2, paragraph 3) as well as we submitted a STROBE checklist as supplementary material, and reported a suitable reference.
Reviewer 2 Report
Body Fat changes almost independently of body weight in older ages, which hides the magnitue of the obesity issue in this age group as to diagnose an individual as obese, WHO recommends the use of BMI. A new algorithm separating the lean mass from fat mass is needed.
P1-L2-3: İt is stated that “ almost 5% each decade after the age of 30 (2), 33 and this may lead to an overall reduction of the latter by nearly 30% between the ages of 34 30 and 70 years” but this math does not seem to work as there are 4 decades between Age-30 and Age-70, which should result in about 20% lean mass loss.
P2-L72-73: Repeating the exact opposite of the inclusion criteria as exclusion criteria is not needed.
P3, L117-118: The power calculations seems to be a post-hoc power calculation. If so, it should be stated as so.
Figure-1: Did the authors check the significance of a quadratic association as the figure provides a hint of it as well? I suggest that the figure be produced with some transparency for the two groups so that the overall distribution for males is not masked by that of females.
P6, L1-2: In these diagnostic models, only the linear effect of BMI was used. Did the authors try the quadratic effect of BMI as well, which I suspect will increase the resulting AUC significantly and the recommended cut-offs can be expressed similarly for such models as well?
P11, L152: The sentence does not read well. Modification recommended
Author Response
P1-L2-3: İt is stated that “ almost 5% each decade after the age of 30 (2), 33 and this may lead to an overall reduction of the latter by nearly 30% between the ages of 34 30 and 70 years” but this math does not seem to work as there are 4 decades between Age-30 and Age-70, which should result in about 20% lean mass loss.
Response: Corrected as suggested (Page 1, paragraph 2).
P2-L72-73: Repeating the exact opposite of the inclusion criteria as exclusion criteria is not needed.
Response: Done as suggested. Removed the opposites (Page 2, paragraph 3).
P3, L117-118: The power calculations seems to be a post-hoc power calculation. If so, it should be stated as so. Response: Done as suggested (Page 3, paragraph 2).
Figure-1: Did the authors check the significance of a quadratic association as the figure provides a hint of it as well? I suggest that the figure be produced with some transparency for the two groups so that the overall distribution for males is not masked by that of females.
Response: Cumulative sum linearity test (CUSUM linearity test) that determinates a linear association based on the distribution of residuals was conducted. For all age and sex groups the CUSUM test confirmed a positive linear association between BF% and BMI (p value >0.05) (Zulle, 2011) (Page 3, paragraph 2). Further, for the purpose of transparency in presenting the scatter plots, as requested by the reviewer, a separate figure was added for the different age and sex groups (Page 5, Fig 1a-f).
P6, L1-2: In these diagnostic models, only the linear effect of BMI was used. Did the authors try the quadratic effect of BMI as well, which I suspect will increase the resulting AUC significantly and the recommended cut-offs can be expressed similarly for such models as well? Response: Given the confirmed linearity, a quadratic association is not needed. However, the quadratic effect of BMI was used for one model (Females between 40-59 years) and the same results were obtained for AUC, sensitivity and specificity as well as the cut off points. Results are shown below and in the attached figure (ROC curve only for revision).
P11, L152: The sentence does not read well. Modification recommended
Response: The statement has been rephrased to appear clear as suggested (Page 11, paragraph 3).

Round 2
Reviewer 1 Report
The authors have made appropriate revises and explanations for each reviewer's comment.